# Evaluating the effectiveness of ensemble voting in improving the accuracy of consensus signals produced by various *DTWA* algorithms from step-current signals generated during nanopore sequencing

**Michael Smith**[1,2]* , **Rachel Chan**[1‡], **Maaz Khurram**[1‡], **Paul M. K. Gordon**[3,4]

**1** Department of Electrical and Software Engineering, Schulich School of Engineering, University of Calgary, Calgary, Alberta, Canada, **2** Department of Radiology, Foothills Medical Centre, University of Calgary, Calgary, Alberta, Canada, **3** Centre for Health Genomics and Informatics, Foothills Medical Centre, University of Calgary, Calgary, Alberta, Canada, **4** Alberta Children's Hospital Research Institute, Calgary, Alberta, Canada

☯ These authors contributed equally to this work.
‡ RC and MK also contributed equally to this work.
* smithmr@ucalgary.ca

## Abstract

Nanopore sequencing device analysis systems simultaneously generate multiple picoamperage current signals representing the passage of DNA or RNA nucleotides ratcheted through a biomolecule nanopore array by motor proteins. Squiggles are a noisy and time-distorted representation of an underlying nucleotide sequence, "gold standard model", due to experimental and algorithmic artefacts. Other research fields use dynamic time warped-space averaging *(DTWA)* algorithms to produce a consensus signal from multiple time-warped sources while preserving key features distorted by standard, linear-averaging approaches. We compared the ability of *DTW* Barycentre averaging (*DBA*), minimize mean (*MM*) and stochastic sub-gradient descent (*SSG*) *DTWA* algorithms to generate a consensus signal from squiggle-space ensembles of RNA molecules *Enolase*, *Sequin R1-71-1* and *Sequin R2-55-3* without knowledge of their associated gold standard model. We propose techniques to identify the leader and distorted squiggle features prior to *DTWA* consensus generation. New visualization and warping-path metrics are introduced to compare consensus signals and the best estimate of the "true" consensus, the study's gold standard model. The *DBA* consensus was the best match to the gold standard for both *Sequin* studies but was outperformed in the *Enolase* study. Given an underlying common characteristic across a squiggle ensemble, we objectively evaluate a novel "voting scheme" that improves the local similarity between the consensus signal and a given fraction of the squiggle ensemble. While the gold standard is not used during voting, the increase in the match of the final voted-on consensus to the underlying *Enolase* and *Sequin* gold standard sequences provides an indirect success measure for the proposed voting procedure in two ways: First is the decreased least squares warped distance between the final consensus and the gold model, and second, the voting generates a final consensus length closer to the known

**Data Availability Statement:** All data, code and demonstration scripts are available at GitHub.com/Nodrogluap/DTWA.

**Funding:** Funding was received from Genome Alberta's Enabling Bioinformatics Solutions Competition (PG) (https://genomealberta.ca; Grant EBS-9). An undergraduate summer research scholarship (MK) and partial publication charges were provided under an Analog Devices' University Ambassadorship Award for Teaching and Research (MS) (https://www.analog.com). No funding from other sources was provided for this project. The funders had no role in study design, data collection and analysis, decision to publish, or preparation of the manuscript.

**Competing interests:** The authors have declared that no competing interests exist.

underlying RNA biomolecule length. The results suggest considerable potential in marrying squiggle analysis and *voted-on DTWA* consensus signals to provide low-noise, low-distortion signals. This will lead to improved accuracy in detecting nucleotides and their deviation model due to chemical modifications (a.k.a. epigenetic information). The proposed combination of ensemble voting and *DTWA* has application in other research fields involving time-distorted, high entropy signals.

## Author summary

Nanopore sequencing devices, essentially a matrix full of microscopic pores, provide an interesting new route in identifying changes in DNA/RNA sequences related to diseases. Biological molecules are sucked down an electrical gradient through the pore while changes in the molecule's electrical characteristics are determined to identify its components. To avoid the sequence information being read as if attached to a rapidly rewound magnetic tape, other biomolecules are introduced to cause the sequence to be ratcheted, rather than free fall, through the pore. However, we are left with an ensemble of pico-amperage nano-signals full of misreads and other experimental distortions. We have demonstrated that it is possible to move dynamic time warped space averaging (*DTWA*) techniques into this high information environment. Consensus signals are generated from multiple noisy signals that are so warped that classical averaging techniques fail. To further improve the quality of the consensus signal, we introduced a new idea in allowing the noisy ensemble of signals as a whole to vote on whether specific *DTWA* consensus components were valid or still a misread. Although areas of further improvement have been identified, the *voted-DTWA* approach already provides cleaner consensus estimates from experimental RNA studies.

This is a *PLOS Computational Biology* Methods paper.

## 1—Introduction

Picoamperage signals are sampled at 3000 or 4000 Hz as the nucleotides of a DNA or RNA molecule are ratcheted by motor proteins through the nanopore biomolecules in a sensor array [1] on a device such as the Oxford Nanopore MinION sequencer [2,3]. Study of nucleotide modifications is critical to understanding many biological regulatory processes. The state of the practice is to analyze these raw current signals using techniques such as black box neural networks [4] e.g., scrappie and bonito (Oxford Nanopore Technologies, Oxford, UK), Hidden Markov Models [5]. A fundamental limitation of these methods is needing a (laborious to derive) large training truth set for any chemical modifications, so the network can detect those nucleotide modifications.

An alternative approach is to process and analyze the streams of stepped current levels (a.k.a. "squiggles" [6]), Fig 1, generated following algorithmic segmentation of the raw current signals. There is a high level of information entropy in the resulting squiggle signal (see zoomed section) because ideally there is a squiggle step-event to base-called ratio of 1:1 between a picoamperage level and a k-mer grouping of nucleotides passing through the nanopore. Other

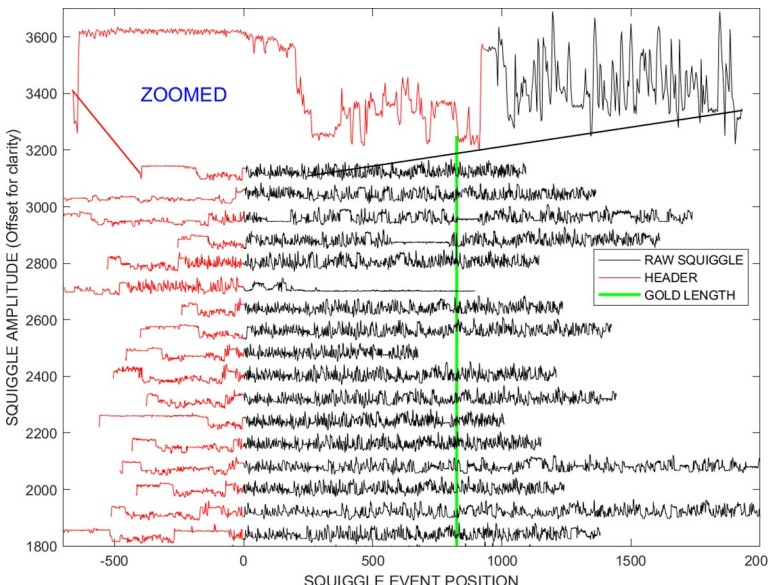

**Fig 1. Identified leaders in the *Sequin R2-55-3* study are shown in red.** There are visually obvious internal distortions in the mainstream body, black, due to factors such as *in-silico* chimeric reads and mis-segmented increased pore dwell times. Identification and removal of very localized errors at the individual k-mer group level is not straight forward because of the high level of information entropy in the signal, see zoomed black section. Ideally there is a squiggle step-event to base-called ratio of 1 between a picoamperage level and a k-mer grouping of nucleotides passing through the nanopore.

deep learning techniques either start with the squiggle [7] after segmentation by the OEM software MinKNOW, or perform the segmentation themselves [8].

In this paper we describe an alternative process which forms a low-noise consensus from a set of distorted squiggles which can be used stand-alone or as part of a preprocessing step for the algorithms that process the squiggle as their input. Our own long-term goal is to generate an improved approach to software that employ statistical white box methods to identify potential chemical nucleotide modifications in existing datasets by identifying non-gaussian distributions in characteristics of ensembles of experimental squiggles.

We believe that our recent preliminary investigations [9,10,11] were the first to propose and evaluate dynamic time warping Barycentre averaging *(DBA)* [12,13] as a potential tool for generating a consensus squiggle from multiple noisy signals from a squiggle ensemble. *DBA* is one of a class of dynamic time warped-space averaging *(DTWA)* algorithms capable of combining an arbitrary number of datasets from multiple sources. This paper is a major extension to those studies and involves a comparison of the *DBA* algorithm with the minimize mean *(MM)* and stochastic sub-gradient descent *(SSG)* algorithms proposed by Schultz and Jain [14] for their relative effectiveness in this new application area of generating a consensus related to the known gold standard available for RNA spike-ins *Enolase*, *Sequin R1-71-1* and *Sequin R2-55-3* [15]

We had hypothesized that as we increased the number of squiggles in the ensemble being averaged, each *DTWA* algorithm's ensemble consensus signal would converge along different paths to produce a similar match to the gold standard squiggle, with consistent differences indicating the presence of possible chemical nucleotide modifications. However, the mean lengths of the consensuses generated from all three algorithms remained systematically longer than the known RNA biomolecular underlying each of the squiggles in the ensemble. A longer length of the consensus signal is a clear indication that any given *DTWA* approach still retains

a large number of duplicated points present in squiggle inputs with segmental duplications. In addition, our simulation studies [10,16] have demonstrated that distorting the gold standard with a large number of segmental duplications according to the Extreme Value Distribution produced the best match to the characteristics of the empirical RNA ensembles.

To remove these unwanted segmental duplications, in the absence of a universal 'perfect' segmentation algorithm for noisy nanopore signals, we propose combining *DTWA* algorithms with an ensemble voting scheme to identify and remove segmental duplicates. This should improve the local similarity between the consensus signal and the majority of the individual noisy squiggle sequences.

The paper is organized as follows. The Methods section outlines the procedure for obtaining the experimental data sets and how to model an appropriate gold standard squiggle. Details are provided of an automated cleaning process to discard damaged and distorted squiggles significantly different from the general ensemble. General descriptions of the three *DTWA* algorithms used to generate an initial consensus and the voting procedure used to remove squiggle distortions remaining in this consensus are provided. The next section discusses the efficiency of the ensemble cleaning process. New quantitative and qualitative success metrics are introduced and used to provide an analysis of the effectiveness of the proposed *DTWA* consensus generating approaches applied to *Enolase* mRNA spike-in and RNA *Sequin v1* Pool A [15] squiggle studies. A comparison of the manner in which the different *DTWA* algorithms converge to a consensus is detailed.

The accuracy and precision of future research using the *voted-on DTWA* consensus are discussed in terms of the assumptions used when developing the voting procedure. The Conclusion section summarizes the advantages of the proposed investigation and outlines the direction of future work needed to resolve the discovered limitations of the combined *DTWA* and voting algorithm in consensus generation in this exciting new application field. The applicability of these results to other signals with very high information entropy are discussed.

## 2—Methods

In this section we first detail the generation of the nanopore sequences which are segmented into step-level squiggles. These are empirically a noisy, stretched and distorted representation of the DNA or RNA present in the stream and must be cleaned of gross distortions before being presented to a *DTWA* algorithm to be processed into a consensus. After detailing key characteristics of the *DTWA* algorithms investigated, we discuss our proposed ensemble voting approach to generate a refined consensus. Fig 2 provides a schematic of each stage of the proposed process.

### 2.1—Data generation and cleaning of multiple noisy squiggle sequences

Four squiggle ensembles were used in this study. A large *Enolase* ensemble was broken into groups of 512 squiggles to allow generation of 20 consensus squiggles to explore the extent to which the consensus changed over an extended acquisition time period. This is in response to an observation [9,10] that later acquired squiggles in an ensemble became longer in length, an effect attributed to an increase in reading errors as the nanopore characteristics, and possibly the sample, degrade. These large ensemble consensus characteristics were compared with those from two smaller and experimentally noisy studies, *Sequin R1-71-1* and *Sequin R2-55-3* ensembles and a small control *Enolase* ensemble, each containing approximately 128 squiggles.

In this paper, the term original equipment manufacturer (OEM) refers to Oxford Nanopore Technology, Oxford, U.K. Data were generated using the SQK-RNA001 direct RNA

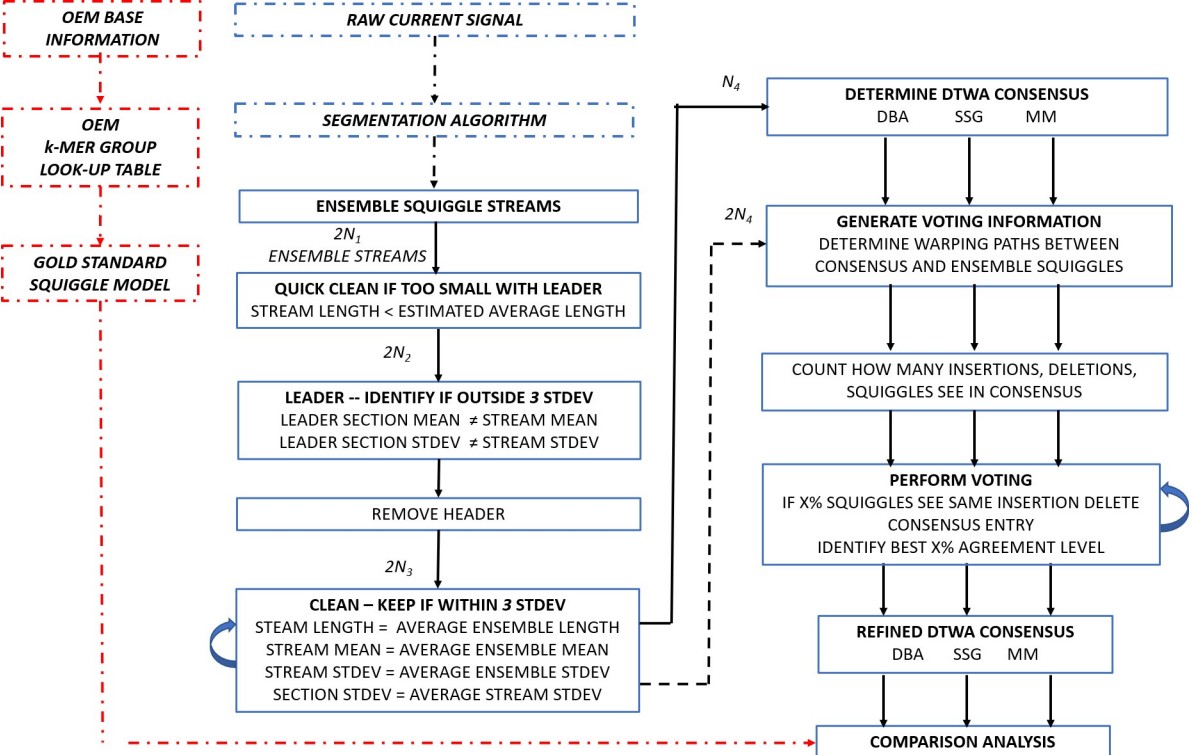

**Fig 2. Flow chart representing the process to generate a refined squiggle consensus by applying an ensemble-wide voting scheme to *DBA*, *SSG* and *MM DTWA* generated consensus signals.** Each cleaning stage reduces the number of streams. While the consensus is based on *DTWA* processing of cleaned data streams, the voting scheme can be based on the agreement with additional streams not involved in generating the consensus experimentally available. Gold standard information is used as part of the final comparison analysis and plays no role in squiggle cleaning, consensus generation or the voting procedure.

sequencing kit and protocol v1.08 with the MinION Mk1B [2,3] sequencer running the OEM MinKNOW device control software (Version 1.11.5). One RNA sample contained only the OEM-provided yeast *Enolase* SGD: ENO2 YHR174W mRNA spike-in. Two others, *R1-71-1* and *R2-55-3*, were supplemented with an RNA *Sequin* v1 Pool A spike-in [15] supplied by Garvan Institute for Medical Research, Sydney, Australia. The raw sampled picoamperage current signals were converted into amperage level squiggles using the open-source current signal to squiggle convertor in *GitHub.com/Nodrogluap/DTWA*.

Oxford Nanopore Technology has performed many experiments in order to provide a k-mer model [17] by 3' end-aligning the known reference nucleotide sequence to expected experimental current levels. This information was used to derive an estimate, i.e. model, of the 'golden' squiggle underlying our experimental signals.

An enzyme-DNA complex is added to the RNA nucleotide stream to guide the RNA through the sensor. While this leader, red lines in Fig 1, is physically essential to perform the experiment, it becomes a variable length artefact that will distort further analysis. Following initial experimentation, we adopted the following procedure for removing the squiggle leaders and distortions [10,16].

- Given that *DTWA* algorithms are designed to match signals with a similar information content, all squiggles with lengths lower than 20% of an average ensemble squiggle length were considered significantly degraded and immediately deleted as gross outliers to speed forming a consensus.

- As can be seen from Fig 1, the characteristics of the leader (red) and the main squiggle stream(black) are different. The leader of the stream was defined as those initial sections of the squiggle with mean and standard deviation metrics more than three standard deviations from the stream's general mean and standard deviation characteristics generated from its trailing quarter.

As detailed in the squiggle simulation study performed by Smith et al., [16], experimental generation of the nucleotide sequences and raw segmentation algorithms produce distortions from many sources including:

1. The uneven production of current steps per unit time due to the stochastic nature of the motor protein driving the steps. This leads to a small increase in nucleotide's dwell time in the nanopore sensor which can be incorrectly interpreted as multiple copies of a single k-mer inserted into the data stream, a.k.a. a segmental duplication or insertion, dashed arrows in Fig 1.

2. Homopolymerism where long chains of multiple identical bases are mistakenly merged by the segmentation algorithm,

3. In-silico chimeric reads where the break between consecutive molecules passing through the sensor is not recognized,

4. Experimental sensor errors and noise generated measuring the current by steric configuration of the nucleotides and

5. Other segmentation algorithm artefacts.

We used the following procedure to reduce the number of these more obvious distorted squiggles prior to consensus generation.

- After removing the leaders, we again deleted squiggles whose length was significantly smaller than the ensemble squiggles average length.

- We removed squiggles whose characteristics were more than 3 standard deviations from the global measures of mean intensity, mean standard deviation and mean sequence length of a given data ensemble. The solid arrows in Fig 1 indicate examples matching this criterium. This matched our hypothesis that the optimum consensus signal would be obtained when applying *DTWA* to noisy sequences which shared the same general characteristics. We chose the looser limit of 3 standard deviations, rather than 2, to quickly provide an initial pruning to remove gross outliers while avoiding having to take into account the specific probability density function of the distortions introduced by the squiggle-generation process [16].

- The previous steps still left squiggles with significant internal distortions. We believe that the majority of these distortions may be associated with chimeric reads, i.e., where full and partial nucleotide sequences are combined prior to passing through the nanopore. To assist in identifying and removing such reads, each squiggle was broken into 10 sections and the standard deviation of squiggle intensity calculated for each section. Squiggles were rejected if the average of the section standard deviations for a squiggle was an outlier, more than 2 standard deviations, for the ensemble's average standard deviation.

## 2.2—DTWA algorithm characteristics

Three different *DTWA* algorithms were applied to the noisy data streams with leaders removed to generate a consensus: the *DTW* barycentre *(DBA)* algorithm [12,13] and the

minimize mean (*MM*) and stochastic sub-gradient descent (*SSG*) algorithms *DTWA* algorithms [14] using MATLAB code provided by their authors.

The following is a summary of [18]; a conference presentation discussing *DTWA* algorithms applied in the new Squiggle environment. All of these *DTWA* techniques rely on the refinement of a consensus using *DTW* alignment aggregation until convergence, i.e., no changes from round to round of *DTW* alignment aggregation. *DBA* is guaranteed to converge on the same consensus for a given set of squiggles because the first (time consuming *Order* $(N^2)$) step of the *DBA* is to find the initial "medoid" or "centroid". This is the squiggle with the smallest sum of squares *DTW* distance to all other squiggles. This medoid is the initial estimate of the consensus and defines the length of final consensus. *DBA* then updates the centroid in "batch" mode, meaning that every squiggle is aligned to the centroid using *DTW*, then the results of all the alignments are summed to update the centroid. This means that the *DBA* process, while starting with a fixed length initial estimate, is not sensitive to the input order of the squiggles.

In contrast the *MM* and *SSG* algorithms run in "incremental" mode, updating the consensus after each *DTW* alignment is performed, and therefore may generate a different consensus depending on the order of the input squiggles. The majorization-minimization (*MM*) mean algorithm is described as an incrementally updated consensus equivalent to *DBA* [14]. The *MM* algorithm assumes that the active component being optimized, the *DTW* distance, is convex and differentiable. To our knowledge, no research has been undertaken to confirm that this assumption holds for high entropy signals such as nanopore squiggles. The sub-gradient mean algorithm (*SSG*), as per its name, assumes that the *DTW* distance optimization can be achieved using a sub-gradient descent method, again not proven to be valid for high entropy signals. As with the *MM DTWA* approach, the *SSG* is also incremental. However, the initial medoid is chosen from a random subsample of all squiggles, implying that solutions will vary from run to run unless a fixed seed is set for generating the random numbers.

## 2.3—Proposed voting procedure

Imperfect segmenting of the raw current signals means each individual squiggle has mistakes introduced by the error mechanisms discussed earlier. The simulation studies [9,16] showed that the squiggle length could grow via insertions, introduction of false bases, or be reduced by deletions, skipped true bases. As will be shown in the Result sections, the final length of the initial *DTWA* consensus is of the order of the average length of the ensemble squiggle. This implies that the *DTWA* algorithm applied to a finite number of squiggles does not recognize and correct all insertions and deletions, so that the insertion and deletion distortions experimentally introduced into the ensemble are preserved in the consensus average.

We propose to use the following procedure to identify the insertions and deletions remaining in the consensus. Consider a *consensus* signal derived from a group of $N_4$ noisy squiggles, Fig 2. The *DTW* metric provides information on the similarity between the *consensus* and the *nth squiggle* in an ensemble after each signal has been individually stretched to minimize the total Euclidian distance (L2 norm) between them [19]. Warps must be introduced to realign sequences whenever data are either missing or repeated in one of the sequences. Applying the *dtw()* algorithm [20] will provide both a similiarity measure, $DTW_{DISTANCE-n}$, and the warping paths, $WP_{CONSENSUS}$ and $WP_{SQUIGGLE-n}$, that best match the consensus and $n^{th}$ *squiggle* from which it was derived.

Our proposed voting procedure to identify spurious insertions and deletions in the consensus uses the following alternative interpretation of the warping elements provided by the *dtw()* algorithm. If a $WP_{SQUIGGLE}$ warped path has multiple entries for a given point in the consensus

signal, it implies that part of squiggle has been stretched to match the consensus signal. If a given proportion $V_{DUPLICATED}$ of the available squiggles in an ensemble vote that they have the same multiple entry, we have assumed that the corresponding consensus location is duplicated and must be deleted.

Our voting procedure is currently limited to removing additional events incorrectly inserted into the consensus rather than identifying and inserting events missing from the consensus, a technically more challenging task. We justify this initial approach since the *x1.7* increase in the experimental squiggle length over the underlying gold standard indicates that over-segmentation of the raw 4000 Hz device signal is a common occurrence. Therefore, additional squiggle events in consensus signals need to be deleted with a far higher probability than the need to add events absent from the consensus signals.

The problem is not associated with determining the lower number of occasions when the *DTWA* fails to give the consensus a base present in the majority, $V_{ABSENT}$, of noisy squiggles. The location of those insertions can be identified by a duplication in the consensus warp path, $WP_{CONSENSUS}$, where the consensus signal has been stretched to match a given squiggle having this missing feature. The issue of "*Why is there a problem of what you insert*?" and the validity of the voted-on consensus after ignoring the issue of insertions is discussed in detail in Section 7.

To provide a comparative study, consensus signals from 20 groups of $N_4 = 256$ squiggles from the large *Enolase* squiggle ensemble were calculated using each *DWTA* approach, Voting can be based on the agreement between the *N* data streams used in forming the consensus or, when sufficient streams are experimentally available, these streams and additional streams not involved in generating the consensus, a total of 512. The voting level, percent of agreement between individual ensemble members, was changed between 100%, high agreement, and 10%, minimal agreement. We would consider our approach successful if all 20 groups showed consistent voting behaviour, i.e., when their final corrected consensus signals showed A) a minimum $DTW_{DISTANCE}$ error measures for similar voting levels and B) having that minimum error associated with a voted reduction in the consensus signal length that more closely matched the underlying RNA molecular length which defines the gold standard length. It is appropriate to expect a close, rather than exact, match to the gold standard characteristics as the purpose of generating a consensus from the noisy ensemble is to identify the epigenetic difference between the ensemble and gold standard.

The two *Sequin* studies were experimentally noisier than the *Enolase* study and only provided a single consensus signal from the original, approximately 110, squiggles after filtering, see Table 1. To provide an equivalent comparative multi-group study, the voting performance of these groups was compared to a consensus derived from a similar small group of *Enolase* squiggles, and to the consensus results from the *Enolase* study involving 20 larger groups.

## 3—Results of leader removal and data cleaning

In other research fields, the *DTWA* consensus is generated from multiple data streams with a loose similarity from many sources. Here, we can make use of the fact that the squiggles are

**Table 1. Comparison of the length characteristics of the original and cleaned data sets.** There is an equivalent *x1.7* length distortion level introduced into all data sets. *To generate a valid cross-comparison, only *130* of the available *7000+ Enolase* squiggles were included in this analysis.

| | Original squiggle (with leader) | | Cleaned squiggle (without leader) | | Gold standard model | |
|---|---|---|---|---|---|---|
| Sequence | # | Mean length | # | Mean length | Length | Segmentation distortion level |
| *ENOLASE** | 130 | 2950 ± 614 | 88 | 2286 ± 248 | 1329 | 1.72 ± 0.19 |
| *SEQUIN R1-71-1* | 116 | 1806 ± 572 | 66 | 1294 ± 242 | 825 | 1.56 ± 0.29 |
| *SEQUIN R2-55-3* | 122 | 1665 ± 595 | 72 | 1252 ± 194 | 782 | 1.60 ± 0.25 |

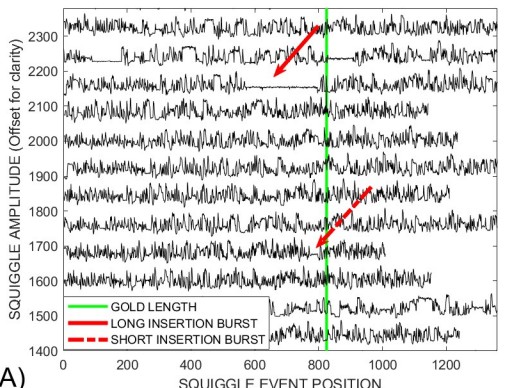
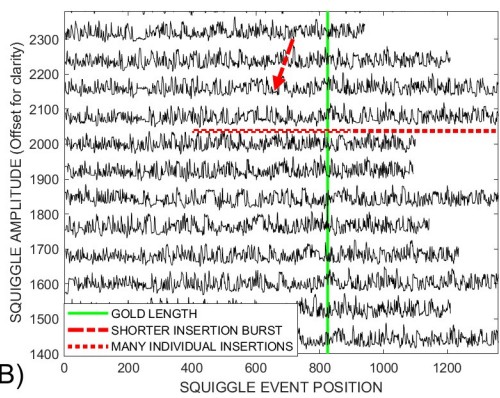

**Fig 3.** A) An initial pruning based on global length, mean and standard deviations of the squiggle ensemble from the *Sequin* study provides a more homogeneous data set than present in Fig 1. There remain obvious long and short insertion distortions (solid and dashed arrows). B) More intensive pruning based on extreme inconsistencies between the local standard deviation statistics of the squiggle to the ensemble statistics leaves only a few obvious insertion distortions (dashed arrow). However, the majority of the streams are significantly longer than the gold standard, indicating the presence of many individual base insertions (dotted line).

essentially a transformation of known information, a genetic gold standard. This means it makes experimental sense to recognize and remove grossly distorted streams from the hundreds available before applying the consensus generating algorithms. Fig 1 showed that the proposed approach shown schematically in Fig 2 correctly identifies the sequence leaders, shown in red, for representative examples from the *Sequin R2-55-3* study containing many distorted squiggles. After leader removal and an initial pruning of the squiggles based on deviations from the ensemble average data length, average data intensity mean and standard deviation, Fig 3A shows that the *Sequin* study still contains significant distorted sequences. Some distortions are long, indicated by the solid arrow near level 2150, and others shorter, dashed arrow near level 1700. Fig 3B shows that the number of internal defects is significantly reduced in number by an additional pruning stage requiring similar intensity standard deviations along the squiggle's length. The visually obvious distortions are few in number, e.g., dashed arrow near level 2175. However, the fact that the majority of streams are significantly longer than the gold standard, green line, indicates the presence of many minor distortions that may impact the *DTWA* algorithms in forming their consensus signals.

Table 1 provides a comparison of the length characteristics of the original and cleaned data sets. Distortions were identified in 34% of the original *Enolase* squiggles, and 40%+ for the *Sequin* squiggles. It was not anticipated that the ratio *mean-cleaned-sequence-length / gold-standard-length* would remain essentially constant at *1.7 ± 0.2* for all three squiggle ensembles, especially as they were produced using different development processes, i.e., natural and synthetic. Considering that these three distinct analyses were performed using the same Oxford Nanopore Technologies MinION flow-cell, we hypothesize that this length distortion level is a property of the cell and segmentation processes. However, we will show that prior knowledge of this ratio plays no role in determining when the *DTWA* or voting processes are considered complete. However, as shown in Fig 2, this knowledge forms one part of determining whether it requires just a particular *DTWA* process, or a combination of *DTWA* and voting, to generate an appropriate final consensus.

A quantitative measure of the level of distortion introduced by both the device sensors and segmentation process can be expressed in terms of the signal-to-noise ratio of the squiggle

length as defined in Eq (1)

$$SNR_{LENGTH} = MEAN(SQUIGGLE\_LENGTH)/STD\_DEV(SQUIGGLE\_LENGTH) \qquad (1)$$

Higher $SNR_{LENGTH}$ ratios indicate more consistent lengths of the members of an ensemble. From Table 1, the small *Enolase* ensemble had $SNR_{LENGTH}$ of *4.8* and *9.2* before and after removing the leaders respectively. This is significantly higher than the $SNR_{LENGTH}$ of the *Sequin* ensembles of *3.0* and *5.0* before and after leader removal. Qualitatively these lower $SNR_{LENGTH}$ values matched the visually obvious increased distortion level of the *Sequin* squiggles compared to the *Enolase* squiggles.

## 4—Metrics to evaluate squiggle consensus generation

To our knowledge, this is the first study of a process that applies *DTWA* algorithms combined with voting to high entropy signals. The final derived consensus representing a global average of the distorted squiggle ensemble should not be expected to do more than resemble the gold standard as there are chemical modifications expected within the experimental squiggle ensemble. However, the gold standard remains the best first approximation to the 'correct' averaging result. In this section we describe a series of qualitative and quantitative tools that compares the gold standard, consensus and squiggle ensemble and can be used to evaluate the performance of both stages of our combined process.

### 4.1—Modified MATLAB *dtw()* display

We post-processed the display generated after calling the MATLAB dynamic time warp algorithm, *dtw(gold, DTWAconsensus)*, [20]. The upper *'original signals'* window, Fig 4, compares the unwarped gold (blue) with the A) *DBA* (black), B) *SSG* (green) and C) *MM* (red) consensus signals. The comparison of the relative consensus distortions is made easier after vertical displacement, rather than the original *dtw()* display's overlap, of the high entropy signals. The length relationship between the *gold* and a given *consensus* is shown in the window title. The increased length of the *consensus* over the *gold* in B) and C) indicates that the *DTWA* consensus characteristic continues to reflect the larger number of additional bases compared to missing bases in the squiggle ensemble as a whole [9,10].

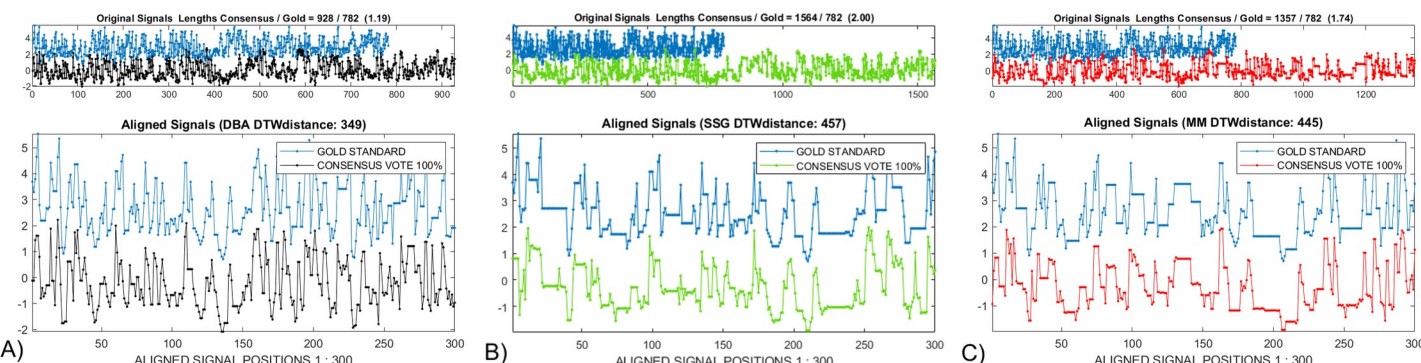

**Fig 4.** A modification of the image from the MATLAB *dtw()* program [20] is useful when empirically comparing the results from the *DTWA* algorithms for the *Sequin R2-55-3*. Entries in the upper window are scaled so that the narrower white band in *A)* shows that the *DBA DTWA* produces a consensus, black, with a length closer to the gold standard than either the B) *SSG*, green, or C) *MM*, red, algorithms. The lower window displays a short, offset version of the aligned signals helping to illustrate the relative level of distortions between the *DTWA* consensuses: B) High for *SSG DTWA*, C) medium for *MM* with A) only the *DBA* consensus signal remaining 20% longer than the gold signal providing an obvious indication of hidden distortions.

The lower window has been resized to allow the warped versions of the *gold* and *consensus* signals to be offset to allow an easier comparison of the similarities and differences between the *gold* and a given *consensus* vertically and between *consensus* signals horizontally. A small portion of the aligned *gold* and *consensus* signals are shown, rather than their original full length, to allow an immediate visualization of significant local differences. Straight portions of the (blue) gold standard warped path are an indication that the *DTWA* process appears to have placed additional bases into the consensus compared to the gold when averaging the noisy ensemble signals. Conversely, a straight portion in the consensus warp path indicates it has been stretched to account for the *DTWA* process apparently leaving bases out when averaging. We interpreted the information in the lower window as providing the following relative levels of distortions between the *DTWA* consensuses: B) High for *SSG DTWA*, C) medium for *MM* with A) only the *DBA* consensus signal remaining 20% longer than the gold signal providing an obvious indication of significant hidden distortions. As discussed in Section 2.2, the *SSG* algorithm will provide different consensus results each time the *DTWA* process is activated unless the *MATLAB* random number generator is seeded, i.e. its initial value is fixed prior to running the code.

### 4.2—Modified warped path displays

Activating MATLAB with the extended command

$$[dtwDistance, goldWarpPath, consensusWarpPath] = dtw(gold, DTWAconsensus) \qquad (2)$$

provides additional information that can be used to compare the gold and consensus signals. A standard approach to explore the relationship between gold and consensus signals is to plot *goldWarpPath* against *consensusWarpPath*, as shown in Fig 5A for the *Enolase* study. However, a direct comparison of features in these paths is not straight forward given that the presence of different length consensus signals leaves these paths vertically offset.

To compensate for the different consensus lengths, we generated a *normalized* warp length display by plotting *goldWarpPath / length(gold)* against *consensusWarpPath / length(consensus)*, Fig 5B. This information is also difficult to interpret as the presence of the *gold* standard underlying every ensemble squiggle implies that the *consensus* will itself be a stretched and dependent version of the *gold* signal. This leaves all normalized warped paths similarly close to the identity line, dotted line.

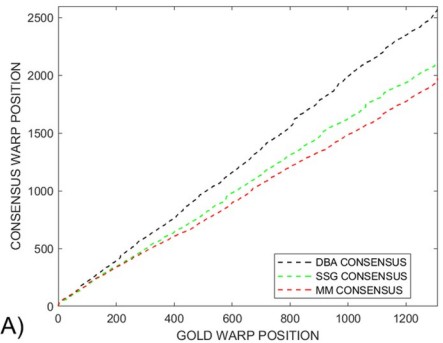 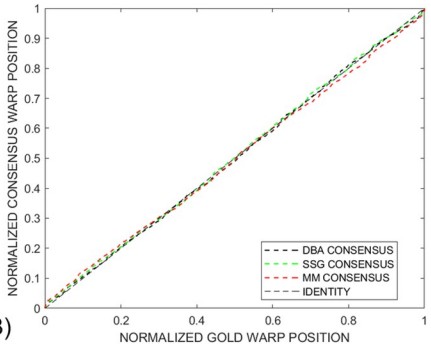 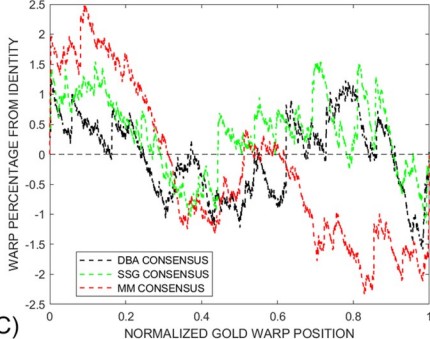

**Fig 5.** A) The standard approach of displaying *dtw* warped plots does not provide a useful route for directly comparing the three *DTWA* consensus signals with each other because of the different consensus warped lengths. B) Normalizing the warped path lengths to one illustrates how close the *DTWA* consensus paths are to the Identity line for the *Enolase* study. This closeness emphasizes the fact that the consensus and original squiggles are essentially stretched versions of the underlying gold standard. C) Plotting the warped path differences from the Identity line shows that all three consensus signals differ in a similar way to the gold signal for the first half of the warp path, with the *DBA* (black) and *SSG* (green) being more similar to each other than with the *MM* consensus (red) in the last part of the warp path.

To enhance the interpretation of *DTWA* consensus differences, we have investigated the use of a *difference from identity (DFI)* warp path display, Fig 5C. We interpret straight sections of the *DFI* path as indicating regions where the consensus and gold signals are noisily similar to each other. Any strong discontinuities in this new path display indicates a sudden dissimilarity between the signals path whose significance should be investigated. All three consensus signals differ from the gold in similar ways in the first part of the warp path but differ in the latter part.

### 4.3—A tool combining warped and unwarped characteristics

Fig 6 shows our approach to combine the best components of the display methods shown in Figs 4 and 5 to assist in the interpretation of results in this new high entropy signal comparison environment. The picture shows the unwarped *gold* (blue) and the *DBA* (black), *SSG* (green) and *MM* (red) consensus signals vertically above each other to better display their local similarities and differences, in particular this approach provides a visual indication of their local amplitude equivalence, or lack there-of, to each other and the underlying gold standard.

Black dotted lines connect corresponding warp path locations in the unwarped signals. The warp-line bars connecting the *gold* and *DBA* consensus being diagonal, rather than vertical, indicates the stretching of this signal relative to the gold signal. The more vertical warp-line bars between the *DBA* and *SSG* consensus signals indicate their similarity, while the more diagonal bars between the *SSG* and *MM* signals again indicate the *MM* consensus signals lower match to the other two consensuses.

### 4.4—Quantitative consensus comparison methods

Quantitative measures are needed to compare the relative accuracy of the three *DTWA* algorithms in producing a consensus, and to determine their respective rate of convergence to an

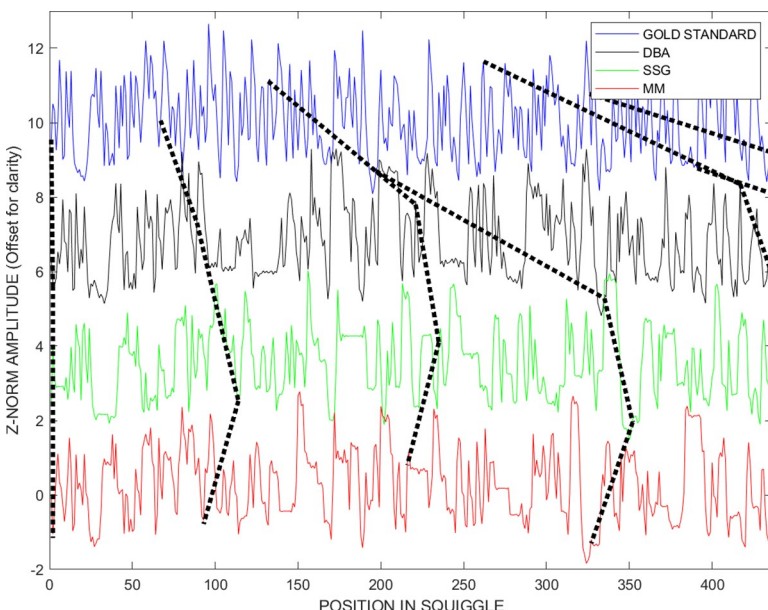

**Fig 6. Displaying the first 400 points of the unwarped gold standard and the three *DTWA Enolase* consensus signals provides an alternative metric combining information from Figs 3 and 4.** This approach allows a visualization of the warp-paths and the extent to which the *DTWA* algorithms retain the high entropy, squiggle amplitude levels characterizing the k-mer groups described in the original ensemble signals. Relative distortions in the consensus are revealed by the unevenly placed, non-vertical orientations of the dashed lines which join points in these un-warped signals identified as having equivalent warped path positions by the *dtw()* algorithm.

improved consensus signal as the number of noisy squiggles that are averaged increases. Previous studies in other research fields comparing the performance of different *DTWA* algorithms [14] have used data sets from multiple sources; each with its own experimental time-warped characteristics. Under this circumstance, an appropriate success metric would be to choose the *DTWA* algorithm that minimizes the normalized mean of the Frechet function *i.e.* producing the lowest mean, Eq 3, and associated standard deviation, Eq 4. of the dynamic time warped (*DTW*) distance, between the consensus signal and the *N* available data streams [14,21]

$$DTWdist_{CONSENSUS \rightarrow ENSEMBLE} = \sum_n DTW(CONSENSUS, STREAM_n)/N \tag{3}$$

$$stdDTWdist_{CONSENSUS \rightarrow ENSEMBLE}$$
$$= \sqrt{\sum_n (DTW(CONSENSUS, STREAM_n) - DTWdist_{CONSENSUS \rightarrow ENSEMBLE})^2/(N-1)} \tag{4}$$

The *DTW* metric provides information on the similarity between a pair of squiggles after each signal has been individually stretched to minimize the total Euclidian distance (L2 norm) between them [19,20]. *DTWA* studies in other fields involve a comparison of the general similarity between signals obtained from multiple sources. This contrasts with this *DTWA* study where it is known that each squiggle is a distortion of a standard squiggle, and a gold-standard approximation of that underlying squiggle can be derived from the known spike-in nucleotides and the OEM provided nucleotide-to-picoamperage mapping table as discussed earlier.

Consensus signals were generated by applying each of the three *DTWA* algorithms, *DBA*, *MM* and *SSG*, to the four ensembles described earlier. To evaluate the difference between a specific consensus and its associated underlying gold standard for each of these twelve studies, we extended the quantitative success metrics proposed in [9] for evaluating *DBA* performance. The metric $DTWdist_{GOLD \rightarrow CONSENSUS}$ involves comparing the aggregate *DTW* distances generated between individual squiggles in the ensemble with the gold and consensus signals. Comparing the metric $DTWdist_{GOLD \rightarrow ENSEMBLE}$ against $DTWdist_{CONSENSUS \rightarrow ENSEMBLE}$ identifies differences in how the gold and consensus signal characterize the noisy data streams. It was assumed that *"smaller is better"* for all *DTW* metrics applied in this study.

We propose new normalized metrics generated by dividing each metric by the underlying gold-standard squiggle length for a given ensemble, *nMetric = Metric/gold_length*, to provide a better tool for comparing several *DTWA* algorithms across multiple DNA and RNA studies with different gold standard characteristics. Each normalized metric takes into account that under equivalent experimental situations, the absolute value of *DTW* distance between the gold standard of a particular study, its consensus and the squiggle ensemble will increase proportionally to the ensemble's gold standard's length.

## 5—*Enolase* results

In this section we qualitatively evaluate the relative performances of the *DTWA* algorithms, with and without voting, in generating a consensus from both a small and a large group of *Enolase* squiggles. The large group analysis provides an indication of what changes may occur in the consensus over a long experiment time, from sources such as sample or nanopore decay. The small group *Enolase* provides a pathway for identifying the expected consistency between the consensus generated during the small *Sequin* study collected over a short time course compared to the longer time course *Enolase* study.

### 5.1—Individual group study

The code runs in four stages, Fig 2.

1. Select *N* streams from squiggle ensemble starting at stream $N_{START}$ and ending at stream $N_{START} + N - 1$.

2. Generate and save cleaned streams if not already stored

3. Generate and save the initial *DBA*, *SSG* and *MM DTWA* consensus signals built from the cleaned streams if not already stored

4. Prepare warping paths between each ensemble member and the consensus. Then determine the number of ensemble squiggles that agree that this warped path location in the consensus is false.

5. In a loop, generate final consensus variants derived from the initial consensus with entries deleted based on agreement voting levels between 100% to 10%. Choose the final consensus from the variants using a success metric.

Fig 7A shows the run times for this research tool when generating the initial *DBA* (black), *SSG* (green) and *MM* (red) *DTWA* consensus signals for ensemble sizes running from 32 to 1024. The code was executed using MATLAB version 2020b on an *AMD Ryzen 5 1600 Six-Core Processor* running at 3.2 GHz with 16 GB internal memory. The *DBA DTWA Order($N^2$)* execution time associated with comparing all *N* cleaned streams with each other when generating the initial centroid is clearly seen. The other *DTWA* algorithms run with *Order(N)* execution times. This is a tool for research and has not been moved onto the final stage of our eXtreme Programming Inspired *(XPI)* approach for software development including refactoring and validation for speed [22]. Planned refactoring will start with parallelization of the many *dtw()* comparisons used in generating the consensus, voting and metrics for analysis.

This paper involves forming consensuses from both the small group *Sequin* studies, 100+ original squiggles, and the large group *Enolase* study, 7000+ original squiggles. Fig 7B compares the changes in the key success metrics as the initial *DTWA* consensus is determined for ensemble grouping from 32 to 1024. The *normalized gold-to-consensus DTW$_{DISTANCE}$*, dotted lines, varies by 10% with the *DBA* metrics being consistently the largest (*smaller-is-better*). The *mean normalized gold-to-ensemble DTW$_{DISTANCE}$* metric, dashed blue line, sits between the *mean normalized consensus-to-ensemble DTW$_{DISTANCE}$* metrics, solid lines, for the *DBA* (black), *SSG* (green) and *MM* (red) *DTWA* consensus signals, with the *DBA* again being largest. This is in contrast with the *Sequin* studies detailed later in Section 6.

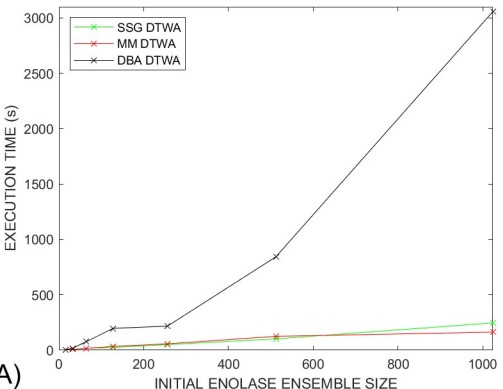
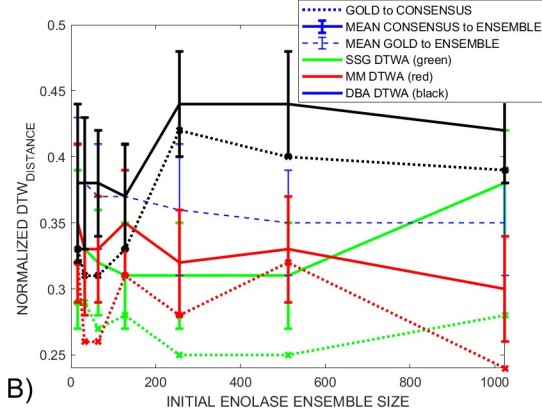

**Fig 7.** A) The *DBA DTWA* execution time is *Order($N^2$)* execution time compared to *Order(N)* for the *SSG* and *MM* algorithms because the initial estimate is generated by compared each nanopore-stream with every other nanopore-stream in the ensemble. B) For the *Enolase* study, the *DBA* difference metrics are larger, *smaller-is-better*, than for the *SSG* and *MM*. differences. This is in contrast with the *Sequin* studies detailed later in Section 6.

Fig 8A, 8B and 8C respectively compare the consensus signals generated by the *DBA*, *SSG* and *MM DTWA* algorithms applied to the *Enolase* study. Unlike the small group *Sequin* study *DBA* result shown in Fig 3, the 549 $DTW_{DISTANCE}$ for the *DBA* consensus is larger than for the 352 and 320 values for *SSG* and *MM* indicating a greater remaining difference from the underlying gold standard. Voting agreement levels were empirically determined to cause the gold length and the *DBA (40%)*, *SSG (45%)* and *MM (43%)* consensus lengths to match. Fig 8D, 8E and 8F compare the final, voted-on consensus signals based on the original *DBA*, *SSG* and *MM* consensus signals. While all voted-on consensuses are visually very similar with significantly reduced $DTW_{DISTANCE}$ measures, the larger *DBA* $DTW_{DISTANCE}$ values indicates a higher level of remaining differences from the gold standard.

Fig 9 provides three difference approaches to compare the warp paths of the final voted-on consensus and gold standard. Fig 9A shows that the different initial warp paths, dashed lines, become very similar after voting, solid lines. This conclusion is also seen in Fig 9B where the voted-on consensus warp paths become close to the Identity line. The *difference from identity (DFI)* warp paths, Fig 9C, show that all three *DTWA* consensus signals deviate no more than 1.5% from the identity line. The *SSG* consensus, green, closely follows the gold standard, +- 0.5% warp path difference, until a sudden deviation is introduced 80% along the warped paths. While the original *SSG* and *DBA* consensus, dashed green and black, are similar along the latter 40% of the warp path, the *DBA* consensus becomes more similar to the *MM* consensus, solid black and red, after voting. The long linear section of the *DBA* consensus difference path from 10% to 99% suggests that the *DBA* and gold signals have a strong similarity except for the strong initial and final deviations which will contribute significantly to the large *DBA*

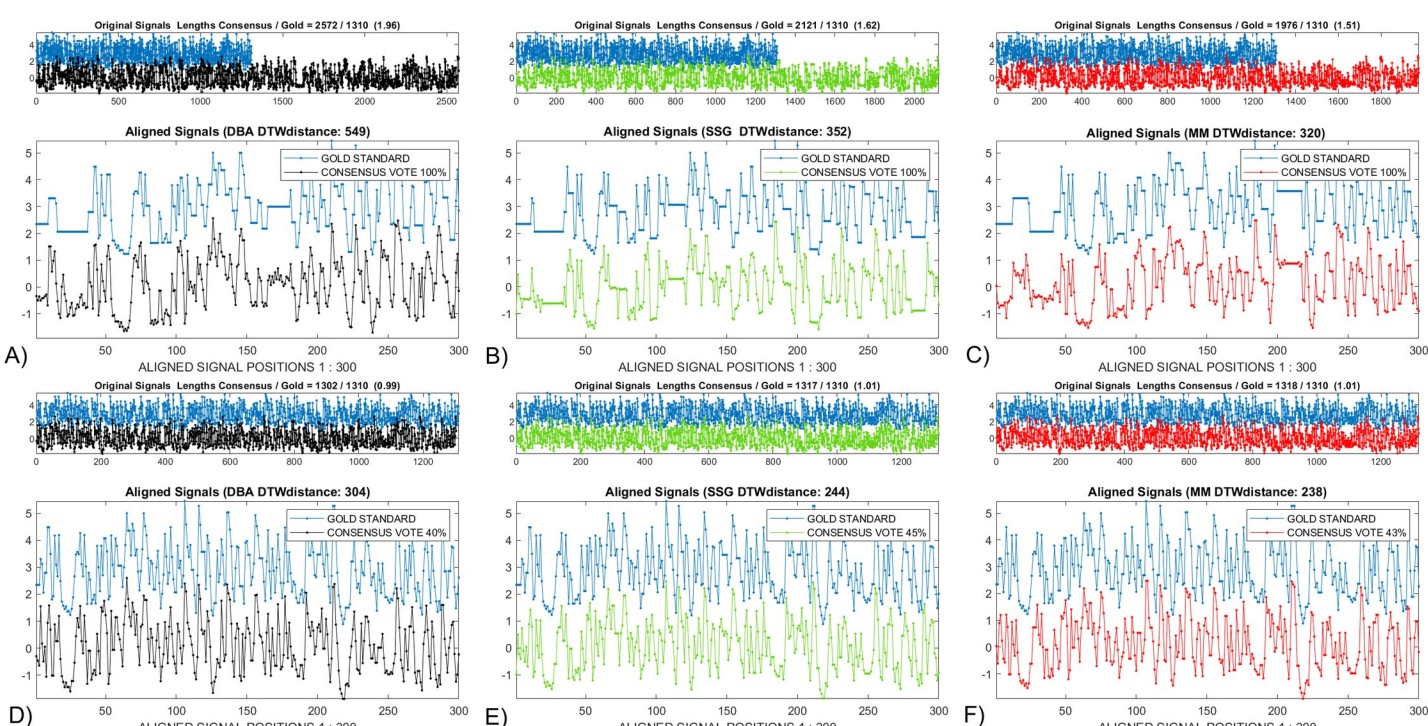

**Fig 8.** The A) *DBA*, B) *SSG* and C) *MM* consensus signals from the *Enolase* study show some similarities and differences before voting. After voting, the D) *DBA*, E) *SSG* and F) *MM* consensus signals appear more visually similar, with only the $DTW_{DISTANCE}$ metric hinting at remaining differences in their relationship to the gold signal.

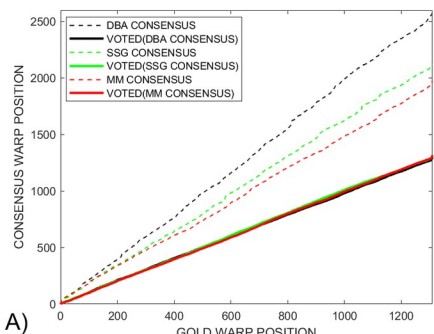 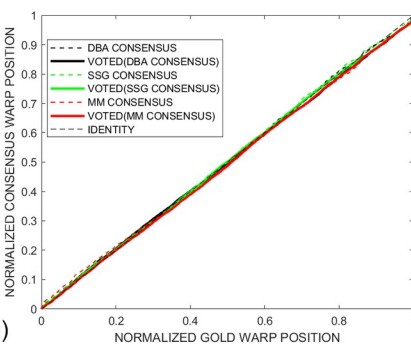 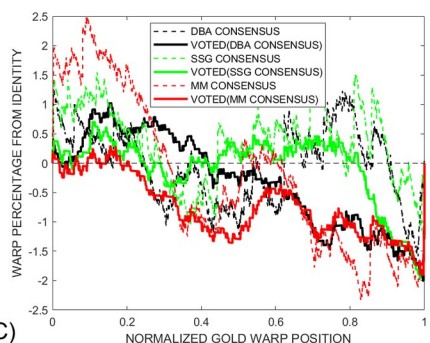

**Fig 9. Dotted and solid lines respectively indicate warping paths before and after voting.** The different *DTWA* consensus warping paths collapse together in both A) the standard and B) normalized warp path displays. C) Linear sections in the *Differences-from-Identity* metric after voting indicate that the *SSG* consensus green, is similar to the gold standard between 20% to 80% of the warp path and the *DBA* consensus, black, is similar between 10% and 99% of the warp path. Several straight sections in the *MM* consensus, red, indicate where this signal is most similar to the gold signal.

$DTW_{DISTANCE}$ measure seen in Fig 7. Several straight sections in the *MM* consensus, red, indicate where this signal is most similar to the gold signal

Fig 10A and 10B shows lines joining equivalent warp-path positions in the last 400 points of the un-warped gold standard and consensus signals before and after voting. The vertical straightening of these warp path indicators in Fig 10B reflects the improved similarity between the three consensus signals, *DBA* (black), *SSG* (green) and *MM* (red), after voting. This alternate approach of representing the similarities and differences between consensus signals clearly shows the reason for the strong distortions near the end of the warp paths in Fig 8C. The last bases, -50 to 0, form a sequence that is common to all consensus signals but absent from the gold standard. We suggest that these distortions be interpreted in terms of the anticipated unreliability of the last squiggle values relative to the golden reference and be discarded. Such distortions are associated with the physical chemistry of the motor protein that drives the RNA through the sensor, and the characteristics of the training set used to derive the *OEM* golden reference discussed in Section 2.

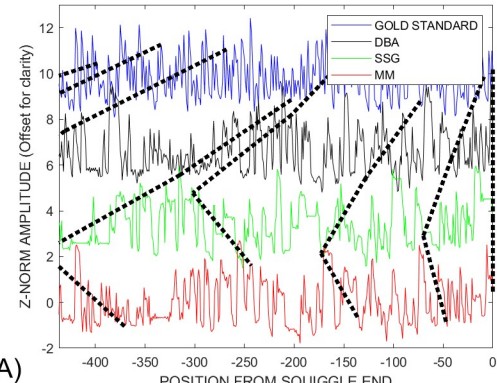 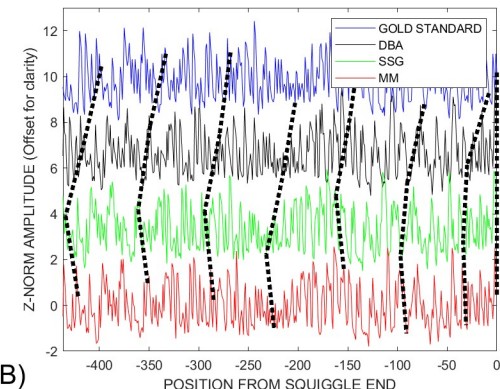

**Fig 10. The dashed and solid lines join points in these un-warped signals with points identified as having equivalent warped path positions by the *DTW* algorithm.** A) There is much less similarity amongst the *Enolase DTWA* consensus signals and to the gold standard before voting than B) after voting. Note that both pictures show the presence of common signals in the last part of all consensus signals that are absent in the gold standard. This difference is responsible for the strong distortions near the end of the *Difference-from-Identity* warped path plot, Fig 8C.

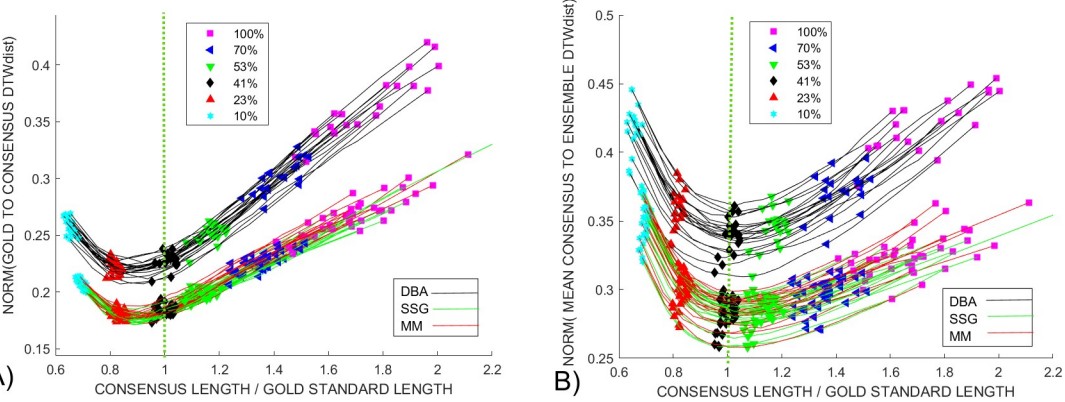

**Fig 11.** A) The change in the normalized $DTW_{DISTANCE}$ between the gold standard and consensus as a function of consensus length is shown for 20 *Enolase* groupings of 512 squiggles as voting level changes from 100% to 10% agreement between noisy squiggles that an insertion occurred. The magenta squares show that initial consensus length, change by 60% during the time taken to perform the experimental study. The *SSG*, *MM* and *DBA* consensuses, green, red and black lines respectively, show similar behaviour after 53% voting agreement, green triangle, reaching a common minimum around 30% agreement.
B) In contrast, a 41% voting agreement generates a minimum, normalized, mean $DTW_{DISTANCE}$ between the consensuses and the original, noisy, ensemble squiggles when the consensus length approaches that of the known gold standard length. This match occurs without the consensus generation and voting process having any prior knowledge of the gold standard characteristics.

## 5.2—Multi group study

We took full advantage of the large *Enolase* ensemble to generate consensus signals from groups of 256 squiggles. The results of applying the voting procedure between 10% and 100% of the noisy squiggles agreeing that there are unwanted features in the consensus signal is shown in Fig 11. The starting lengths, magenta squares marked 100%, of the initial consensus from the *DBA* (black), *SSG* (green) and *MM* (red) *DTWA* algorithms range widely; from *x1.4* to *x2.1* of the length of the gold standard, dotted green line, known to underlie each squiggle in the ensemble. However, the changes in all consensus lengths start to follow a similar pattern by the time 75% of the squiggles, black triangle, are in agreement regarding the level of consensus distortions. All consensus signals show a global normalized $DTW_{DISTANCE}$ minimum between them and the gold standard when their length is approximately 90% of the gold standard, Fig 11A. This minimum is consistently lower (smaller is best) for the *SSG* and *MM* consensus than for the *DBA*, echoing the same differences between the voted-on consensus signals from the smaller *Enolase* grouping in Fig 7D–7F.

Fig 11B shows a weaker global minimum for the mean normalized $DTW_{DISTANCE}$ between each consensus and their respective noisy ensembles. This global minimum consistently appears when 41%, black diamond, of the noisy squiggles agree that the consensus is distorted at a specific location and when the consensus length approximates the gold standard length rather than being shorter as occurs in Fig 11A. From this analysis we conclude that the *voted-on DTWA* consensus better represents the average characteristics of the ensemble of noisy squiggles than the characteristics of the gold standard model.

## 6—*Sequin* consensus study

In this section we report a comparison of the characteristics of the voted-on consensus signals generated for the two *Sequin* studies. Experimental issues meant that only a single group of less than 120 cleaned squiggle was available from either *Sequin* ensembles. A small grouping of the *Enolase* ensemble was included in the study to provide a link to the results of the multiple group study in Section 5.2.

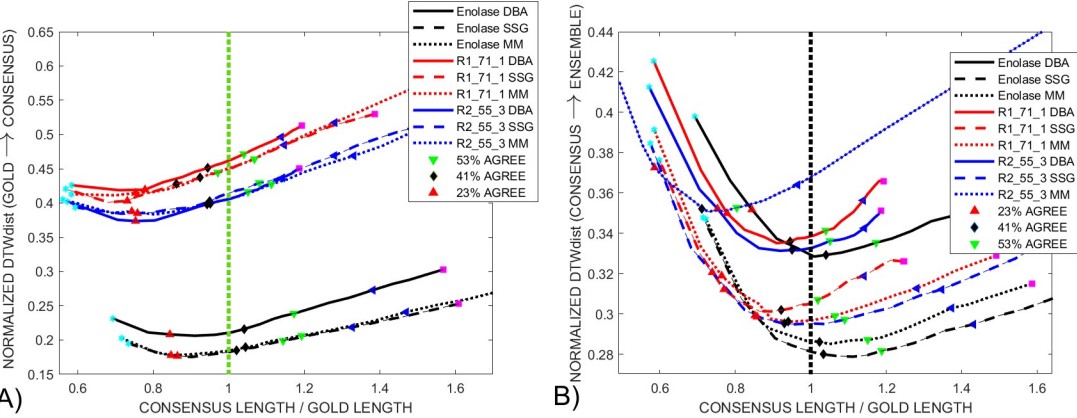

**Fig 12.** A) A comparison of the $DTW_{DISTANCE}$ between the gold standard and consensus for a single 128 squiggle grouping from the *Enolase* (black), *Sequin R1-71-1* (red) and *Sequin R2-55-3* (green) studies using the *DBA*, solid line, *SSG*, dashed line, and *MM* *DTWA* algorithms. All *Sequin* $DTW_{DISTANCE}$ minima occur in the 23% - 30% voting agreement range, lower than for the control *Enolase* study. B) Again, plots of the mean normalized $DTW_{DISTANCE}$ between the voted-on consensus and its ensemble as whole all show a minimum close to their respective, and different, gold standard length despite using a *DTWA* consensus generated from fewer, and for the *Sequin* studies, noisy squiggles.

Fig 12A shows the changes in normalized $DTW_{DISTANCE}$ between the *Sequin R1-71-1* (red), *R2-55-3* (blue) and *Enolase* (black) voted-on consensus signals and their respective gold standards as the level of agreement required between the noisy squiggles in the full ensemble that a duplication in the consensus was present changed between 100% agreement down to 10%. The normalized $DTW_{DISTANCE}$ for the *Sequin* studies are similar to each other, but higher than that of the *Enolase* study. The behaviour for the small group *Enolase* study, black line, has a global minimum when the *consensus length / gold standard length = 0.9*, echoing the results of the multi-group *Enolase* study in Fig 11A. In contrast, the *Sequin* studies show a global normalized $DTW_{DISTANCE}$ minimum when the ratio of their consensus to gold standard lengths are close to 0.75. Fig 12B shows that the mean normalized $DTW_{DISTANCE}$ between the consensuses and their respective ensemble shows a minimum when the *consensus length / gold standard length* approximates 1.0. As with the multi-group *Enolase* study, Fig 11, this is again confirmation that the consensus represents an average of the experimental ensemble with insertions and distortions present in individual squiggles and is not intended to be an accurate and direct representation of the gold standard model. We currently offer no explanation of what characteristics of the *MM DTWA* consensus generated in the *Sequin-R2-55-3* study, dotted blue line, makes it so similar to the other consensus signals when compared to the gold standard, Fig 12A, yet obviously inconsistent in its normalized, mean $DTW_{DISTANCE}$ minimum behaviour when compared to the ensemble, Fig 12B. This shows that while the *MM*, *DBA* and *SSG DTWA* algorithms generally produce similar results, specific characteristics of data and consensus initialization may result in different results.

Fig 13 compares the three variants of the warp path display for a small control group *Enolase* study, Column 1, against the equivalently sized *Sequin R1-71-1*, Column 2, and *R2-53-3*, Column 3, study groups. The standard warp-path comparison between the gold and consensus, Row 1, show that all consensuses converge to similar values after voting, becoming close to the Identity line of the normalized warp-paths, Row 2. The *Difference-from-Identity-Line* plots reveal interesting insights into some *DWTA* behaviour. The random *SSG* initialization results in very different initial consensus signals, dashed green lines when applied to a large *Enolase* ensemble, Fig 8C, and a small *Enolase* ensemble, Fig 13G. However, voting makes the

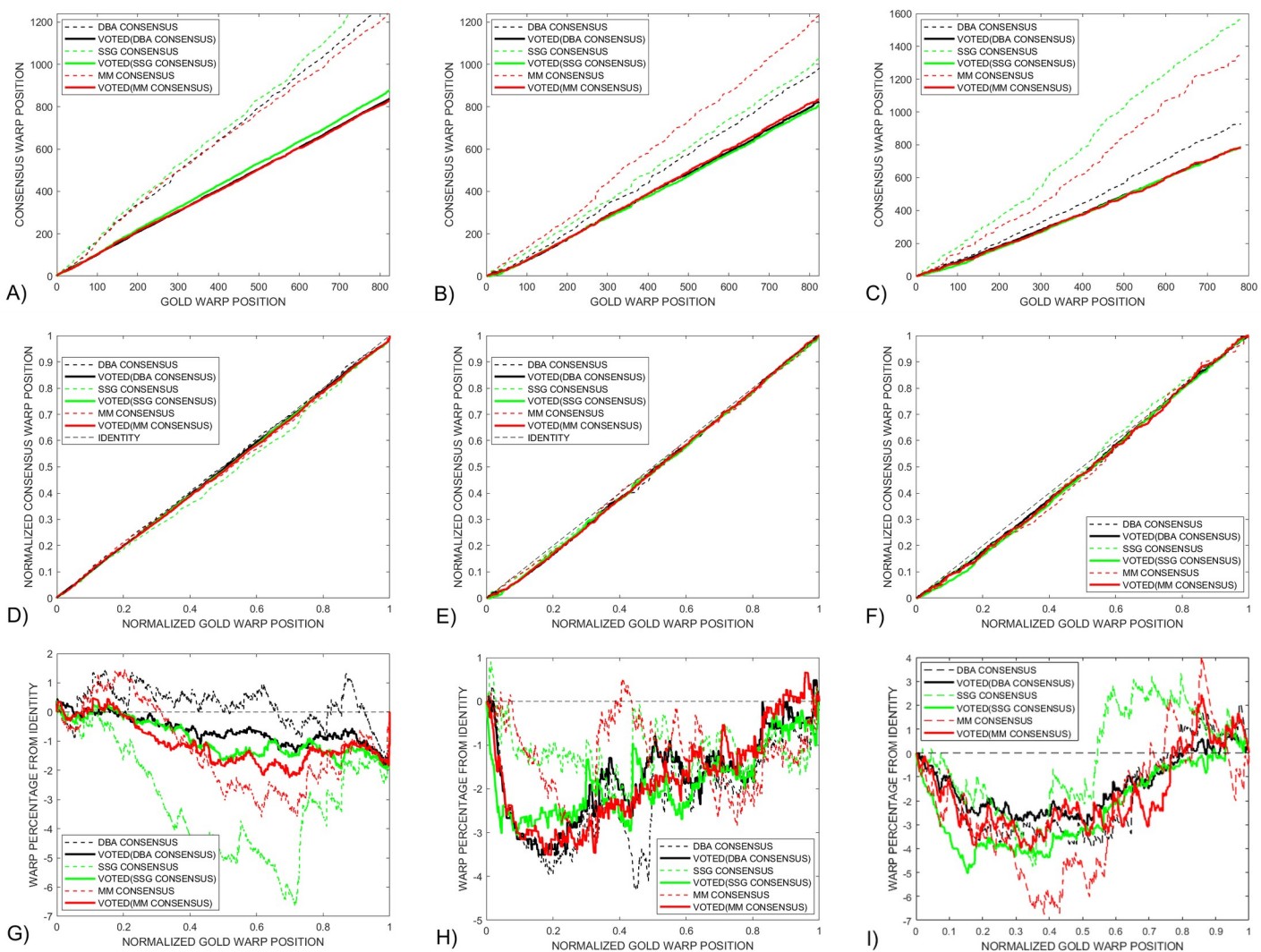

**Fig 13. A comparison is made between various warp-path metrics for the *Enolase*, column1, *Sequin R1-71-1*, column 2, and *Sequin R2-55-3*, column 3, with normalized warp position calculated from their respective gold lengths.** The normal warp path display, Row 1, shows how the *DBA*, black, *SSG*, green, and *MM*, red, signals all gain more similar lengths upon voting. These changes are reflected in the normalized paths, Row 2, which show a drop in deviation from the Identity path after voting. The *Difference-from-Identity* warp paths, Row 3, shows that the three *DTWA* consensus signals become similar after voting.

consensus signals more similar to each other. Straight sections of the *Enolase* warp-path plots. Fig 13G indicate that the consensus has much similarity with the gold standard over essentially the total warp path except for the final few per cent. This contrasts with the *Sequin R1-71-1* and *R2-53-3* studies, Fig 13H and 13I, which consist of two straight difference warped-path with sudden changes occurring at the 20% and 50% positions along the normalized warp paths. We consider the full interpretation of the cause of these changes in this new consensus metric beyond the scope of this paper.

Fig 14A and 14B respectively compare the last base values of the unwarped *Sequin R1-71-1* and *R2-55-3* DWTA consensus signals with their own unwarped gold standard before voting. The initial *DBA* (black) and the *SSG* (green) consensuses show similarity to each other in the *Sequin R1-71-1* study while both are obvious distorted versions of the gold standard. This contrasts with the *Sequin R2-55-3* study where the original *DBA* consensus shows significant resemblance to the gold standard, presumably as the result of its time-consuming '*find-the*

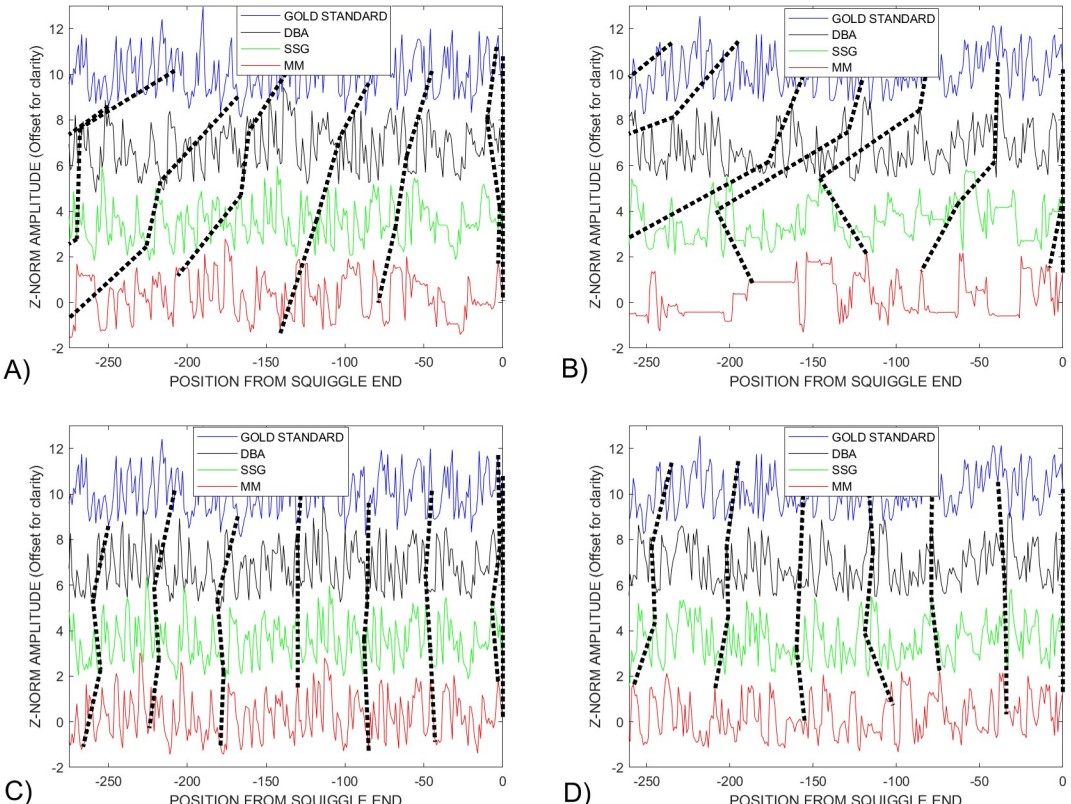

**Fig 14.** The original, unwarped *SSG*, green, *MM*, red, and *DBA*, black consensus signals for the A) *Sequin R1-71-1* and *B) R2-55-3* studies show significantly more distortions, non-vertical dashed lines between equivalent warp points, than were present in the *Enolase* study, Fig 9A. After voting, the consensus signals within the C) *R1-71-1* and D) *R2-55-3* studies become more equivalent to each other as shown by the more vertical dashed lines connecting equivalent warp positions within the unwarped signals.

*most-consistent squiggle'* initialization applied to this small, noisy, data set. The vertical positioning of the dotted lines joining identified similar warp path location clearly shows that the final voted-on consensus are similar to each other, and the gold standard, in both the *Sequin R1-71-1* and *R2-55-3* studies, Fig 14C and 14D respectively. We need further study to understand why the last sections of the synthetic *Sequin* consensus signals are similar to the gold standard in contrast to the results seen with the end segments of the naturally derived *Enolase* studies, Fig 9B.

## 7—Accuracy and precision of the *voted-on DTWA* consensus

In this Squiggle context, precision is related to the repeatability of the *DTWA* and voting process. Unless the starting point, initial squiggle, or some other step is chosen randomly, the *DTWA* is a mathematical process and that always lead to the same consensus signal for a given *DDBA*, *SSG* or *MM* application. However, consensus differences are expected for a number of reasons. Smith et al. [10] showed that forming ensembles from groups of squiggles of similar length leads to a different consensus. In addition, our multi-group study, Fig 11A, shows that the initial consensus signal length, magenta square, and hence other characteristics, changes when comparing squiggle groups of random lengths from the same ensemble.

Voting is also a mathematical process. Fig 11A illustrates that voting leads to similar voted-on consensuses characteristics across different *DTWA* methods and squiggle groups regardless of the initial consensus length. It is reasonable to assume that the combined *DTWA* and voting processes described in this paper are precise in terms of repeatability.

A decision on accuracy must be inferred, rather than measured, as the 'correct result' of applying *DTWA* techniques is not to generate an accurate representation of an unknown gold standard by combining multiple noisy signals. Instead, as Fig 11B shows, the purpose is to generate a consensus signal for use as a representative of the average ensemble to determine systematic differences, possible chemical nucleotide modifications, between it and a minority of individual signals in the ensemble that are different from the majority.

Figs 11 and 12 together provide clear decisions for the existence of global minimum $DTW_{DISTANCE}$ measures when approximately a 40% - 45% majority of four different ensembles agree on the presence of distortions in their respective consensus. To proceed with using such consensus in future studies, we must answer several questions regarding consensus generation in this new squiggle context–"*What are the unsatisfied 55% - 60% voters unhappy about with the current analysis?*" and "*Is their unhappiness significantly influencing the final result?*" Finally, "*Is it useful to address their concerns in the short term?*"

Earlier, we discussed how applying the *dtw()* algorithm to compare a *DTWA* consensus signal estimate and an individual squiggle returned two warped paths. We have developed a voting method that relied on additional entries in individual squiggle warp paths, $WP_{SQUIGGLE-n}$, indicating that bases should be deleted from the consensus. We will now provide an argument that ignoring the fact that additional entries in the consensus warp path, $WP_{CONSENSUS}$, indicate missing entries in the consensus does not lead to a first order bias in the *deletion-from-consensus* voting process.

Assume that 5 in 100 squiggles indicate a missing consensus base that must be added. Then to achieve a *x1.7* average ensemble length ratio to the gold standard increase requires that at least 75 in 100 squiggles indicate a deletion should be performed. Only when deletion voting has caused the consensus to reach approximately the gold standard length will there be roughly equal numbers of squiggles, 5 in 100, requesting either a deletion or an insertion. We also argue that even continuing to ignore the requests for insertions will not introduce more than a second-order *delete-from-consensus* bias as the need to insert and delete bases will not normally occur at the same location unless both signals are so grossly distorted that it impacts the convergence of the *dtw()*.

However, there is an argument that a second order bias will be present since continuing to ignore minority insertion requests will have some impact on the consensus characteristics and can be expected to increase the overall $DTW_{DISTANCE}$ metric. Duplication of the basics of our deletion process would, in principle, allow a determination from the consensus warp path, $WP_{CONSENSUS}$, returned by the *dtw()* algorithm of when an ensemble vote is high enough to indicate an insertion should occur. However, "*what to consider as a valid insert*" does not have an immediately obvious answer.

We initially felt that inserting the average of the value at the ensemble warp path location would be acceptable given the presence of experimental noise. However, in the presence of chemical nucleotide modifications, a.k.a. epigenetic changes, we would expect that the distribution amongst ensemble squiggle values would show significant deviations from the normal distribution. Thus, inserting an average value into the consensus without a more in-depth analysis of squiggle distributions might impact the evaluation of these changes, i.e., insert a false negative indicating the absence of a modification. We concluded that adding a valid insertion into the consensus should only occur when we moved onto the next stage of our project,

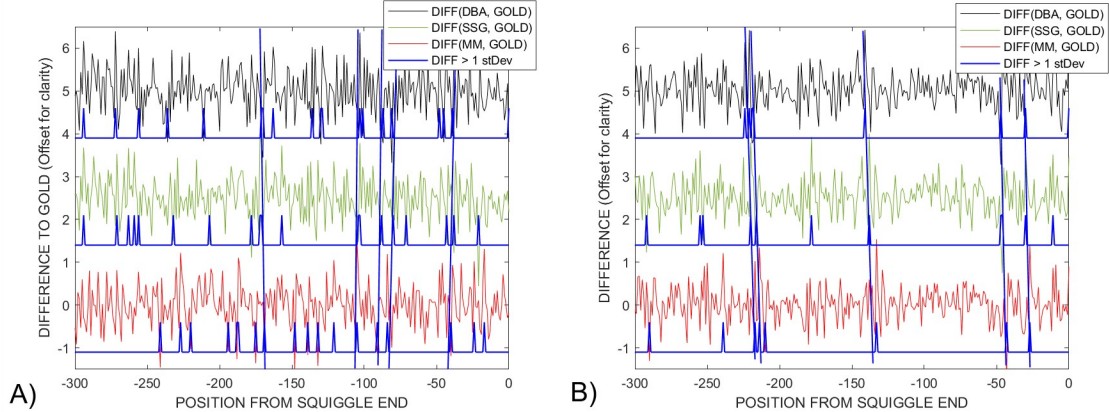

**Fig 15.** Comparison of the differences between the amplitudes of the warped gold and *DBA* (black), *SSG* (green) and *MM* (red) *voted-on DTWA* consensus signals for *Sequin* A) *R1-71-1* and B) *R2-55-3* studies. The short blue spikes indicate where higher than average differences exist between the study's gold standard and a consensus signal, many larger differences being common across all consensuses, blue lines. However, closer examination of all differences show that they cannot be represented as experimental produced gaussian deviations around a mean, but are equivalent, small systematic differences common across all *DTWA* consensuses.

using voting information combined with recursive analysis of amplitude variation and difference distributions to determine the presence of these modifications.

While beyond the immediate scope of this paper, we undertook a preliminary investigation of a related issue–*"Do the results of ensemble DTWA-voting already make it possible to see significant differences common across the three different DTWA approaches between a voted-on consensus from experimental data and the gold signal?"* In Fig 15 we have plotted the difference between the gold signal and the three voted-on consensus signals at each location along their warp-paths. Making an initial assumption that overall differences would be due to random experimental noise, we identified a number of common locations, blue lines, where higher than average differences were seen by each of the *DTWA* consensus signals and the gold standard in both A) *Sequin R1-71-1* and B) *Sequin R2-55-3* studies.

However, closer examination of all 6 signals shows that our initial assumption that the differences would be noise-like with a normal distribution around a zero mean is invalid. Future work is required to understand why these is such a high level of similarity in all the differences between the gold standard and the three consensus signals generated by independent and significantly different *DTWA* processes in two different *Sequin* studies each with its own gold standard.

One consideration is that the similarity may be related to minor, but systematic, deviations from the existing *OEM*-provided k-mer current values when we generated the gold standard model for our particular experiment. This concern would have no impact on the success of our current combined *DTWA* and voting process which does not make use of this generated gold standard during the voting process.

A possible second conjecture is related to the repeated application of multiple *dtw()* steps when generating a *DTWA* consensus. In other research fields, the signals are typically from such disparate sources that an increased level in the similarity of the signal amplitudes must be imposed by z-normalizing each signal, i.e., removing the mean and dividing by a signal's standard deviation. However, the squiggle signals are inherently from the same source, so in principle they should already have similar amplitudes and standard deviations. Thus, any self-normalization approach, e.g., z-norm or median median-absolute-deviation (*med-MAD*), will

effectively be applying different scaling factors based on some standard property to each squiggle reducing, rather than increasing, their inherent similarity; thus potentially introducing a systematic error when applying the *dtw()*. We are currently investigating whether changes in conditions during an experiment, e.g., sample or nanopore aging, make some form of normalization necessary, and any systematic error introduced into the consensus accepted.

## 8—Conclusion

The ratcheting of nucleotides of an RNA or DNA molecule through the biomolecule pores of a nanopore sequencer generates picoamperage current signals. These nanopore signals are segmented into step-current levels, squiggles, related to particular nucleotide groupings. However, these squiggles are consistently longer than the known source RNA length when using the OEM segmentation algorithm based on changes in current mean and standard deviation over a time window. This stretched signal-events-to-base-calls ratio indicates multiple spurious events within each data stream.

We have investigated the effectiveness of *DTW* Barycentre Averaging (*DBA*), the Minimize Mean algorithm (*MM*) and Stochastic Sub-Gradient descent algorithm (*SSG*) dynamic time warping averaging *(DTWA)* algorithms in producing a consensus signal from multiple noisy squiggles. The algorithms showed different properties when applied to three experimental samples, *Enolase* mRNA spike-in (natural) and two studies using *R1-71-1* and *R2-55-3* (synthetic) from the RNA *Sequin v1 Pool A*. Generated gold standard models were treated as 'best first order estimates' when analyzing the success of the consensus generation

The initial squiggle cleaning process to remove uncharacteristic sequences revealed that the nanopore sequencer and associated segmentation algorithm consistently introduced distortions causing an ~*x1.7* increase in sequence length compared to the underlying gold standard. The initial *SSG* and *MM DTWA* consensuses had the lower, *smaller-is-better*, $DTW_{DISTANCE}$ success metrics for the larger *Enolase* study but were outperformed by the *DBA DTWA* consensus on both smaller *Sequin* studies. New visualization and warp-path comparison techniques were proposed in this new environment where one of the signals being compared, the *consensus*, is generated from an ensemble whose individual members inherits the majority of the characteristics of the second signal, the *gold-standard* model.

The increased length of the initial consensus signals compared to the gold indicated the *DTWA* averaging cause a considerable retention of the distortions present in individual squiggles. We have proposed a post-processing procedure where a certain majority of the noisy individual squiggles vote (based on warping path repeats) whether there are false additions present in the consensus signals. Three experimental studies were investigated: a large *Enolase* mRNA spike-in squiggle ensemble and two smaller, noisier, *R1-71-1* and *R2-55-3* ensembles supplemented with RNA *Sequin v1* Pool A. We demonstrated that upon voting, the length of the consensus signals was decreased, and the *voted-on DTWA* consensus became a better match to the ensemble as a whole than did the gold signal underlying individual members of the ensemble. The success measures for the voted-on consensus again showed *DBA DTWA* better for *Sequin* studies, *SSG* and *MM DTWA* better for the *Enolase* study,

We believe that there is considerable potential in applying *voted-on DTWA* algorithms in this new application area, including the identification of chemical nucleotide modifications present in the experimental consensus signal and not in the gold standard. Our future work includes developing new approaches variants that combines the best features of each of the existing *DTWA* algorithms, upgrading the voting procedure to identify where features are missing in the consensus and the more difficult task of proposing a valid correction. The squiggles have very high information entropy because of the 1:1 relationship between the amperage

levels and the underlying DNA/RNA molecule being ratcheted through the sensor. We expect that the results of combining an ensemble averaging algorithm with a voting procedure could be applicable in other fields where signals have similar high-entropy characteristics.

Code, data files together with scripts to convert various data file formats into the simple file format used by this tool can be found at *GitHub.com/Nodrogluap/DTWA*.

## Acknowledgments

The authors wish to thank Drs. Raymond Tellier and Kanti Pabbaraju from the Alberta Provincial Laboratory for Public Health for performing the nanopore device experiments.

## Author Contributions

**Conceptualization:** Michael Smith, Paul M. K. Gordon.

**Data curation:** Michael Smith, Rachel Chan, Maaz Khurram, Paul M. K. Gordon.

**Methodology:** Michael Smith, Paul M. K. Gordon.

**Resources:** Paul M. K. Gordon.

**Software:** Michael Smith, Rachel Chan, Maaz Khurram, Paul M. K. Gordon.

**Supervision:** Michael Smith.

**Writing – original draft:** Michael Smith, Paul M. K. Gordon.

**Writing – review & editing:** Michael Smith, Paul M. K. Gordon.

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
