## [Decision Letter · Decision Letter 0]

29 May 2021

Dear Dr. Smith,

Thank you very much for submitting your manuscript "Comparing the effectiveness of several dynamic time warped space averaging (DTWA) algorithms combined with ensemble voting to generate improved consensus signals from nucleotide sequences produced by a nanopore device" for consideration at PLOS Computational Biology.

As with all papers reviewed by the journal, your manuscript was reviewed by members of the editorial board and by several independent reviewers. In light of the reviews (below this email), we would like to invite the resubmission of a significantly-revised version that takes into account the reviewers' comments.

The most pressing issues raised by reviewers include are related to the details about the data and software, data and code availability to reproduce the results, and code access to be able to apply the proposed methods to other datasets. I hope the other comments from the reviewers are also useful to make your work more accessible and relevant to others. Thank you.

We cannot make any decision about publication until we have seen the revised manuscript and your response to the reviewers' comments. Your revised manuscript is also likely to be sent to reviewers for further evaluation.

Sincerely,

Eduardo Eyras, PhD

Guest Editor

PLOS Computational Biology

Ilya Ioshikhes

Deputy Editor

PLOS Computational Biology

One of the most pressing issues raised by reviewers is the details about the data and software, data and code availability to reproduce the results, and code access to be able to apply the proposed methods to other datasets. I hope the other comments from the reviewers are also useful to make your work more accessible and relevant to others. Thank you.

Reviewer's Responses to Questions

**Comments to the Authors:**

Reviewer #1: The Authors present a comparison of several DTWA algorithms, along with a method for ensemble voting to better match a consensus signal with a "gold standard" signal generated from the sequence of synthetic controls. While the ensemble voting and comparison methods themselves appear sound, and the next steps in using this method to detect modified bases are indeed exciting, and I have confidence will probably work quite well, I found it difficult to review and validate.

My most pressing issues with the current submission, is the lack of raw data and which software (and versions) were used to generate the "gold standard" squiggles, and read event tables. I checked the cited publications, and they also did not mention this information (which should have been picked up before now!). Without this information and data, the methods presented are not reproducible, and therefore do no meet the publication requirements. It also makes it difficult for me to review some of the finer points of the methods. The segmentation algorithms and models used in basecallers have changed many times over the years. These have a direct impact on the results of the methods presented, and their further use by the Author's peers. Furthermore, the code provided as part of the package for "data" is from a different publication, and no code for the methods presented have been provided. This is simply unacceptable for a computationally based manuscript. At the very least these issues should be resolved.

I go into more detail on these points as well as others in my review below to help speed up peer review in the future, but as it stands, the work is not reproducible. I do hope the authors are not discouraged by my review, as I do think the direction of the work is interesting, exciting, and publishable with work on the areas mentioned.

Review:

The title:

"generate improved consensus signals from nucleotide sequences produced by a nanopore device"

The consensus signals are not being produced from nucleotide sequences, they are being produced from the raw signal segmented into events from nanopore sequencing reads.

Fig 1. Image is quite low in resolution (as are all the figures)

Leader sequence difference to the rest of the RNA signal is not easily resolved with the lack of height in each signal. It may help the reader understand how the leader/DNA adapter was detected and trimmed from the description if they can see the striking difference in signal on either side of the PolyA tail.

75: "The state of the practice is to use black box neural networks (5) or hidden Markov models (6) to turn the

raw current signal into nucleotide sequences."

This isn't entirely correct. There are many examples, including from ONT, of event segmentation methods.

for example, the segmentation code from https://github.com/nanoporetech/scrappie is freely available.

Also, the latest basecallers (bonito for example), don't use event tables as as such, working from the raw signal using 1D.

There are also other methods of signal segmentation, such as that used by cwDTW, or any number of open source basecallers (Nanocall, DeepNano...)

2.1 Data generation

Please note the MinKNOW version as well as the "OEM provided current to signal squiggle converter" name and version.

Presumably you are referring to Guppy, although I can't tell how old this RNA001 data is, so it could even be Albacore.

Either way, the metadata for version information can be found in the metadata of the .fast5 files.

The reason this is important, is because the algorithms for signal segmentation have changed a number of times, and without knowing the particulars, your methods are not reproducible.

On that note, the only data provided seems to be flat .txt files with the (i think) event values, with no readID or other identifying features. fast5 files containing the raw signal data to work with in order to generate these event tables, should be included. Again, this is to ensure the methods are reproducible.

Further to this, the original OEM kmer models are not included, and also cannot be derived, for as previously mentioned, no software or version information is given. Only a "gold standard" file is produced. This makes the gold standard generation not reproducible as well.

The base sequences of the particular Sequin and Enolase spike-in controls used should be provided as a supplementary, as this is what is used to produce the "gold standard" from the kmer models.

138-143: The files containing the code which does this such as "RemoveHeader_10Jan2019.m" should be referred to. This applies for all sections that refer to computational methods. For reproducibility, the user should be able to retrace the Authors steps.

168: Wouldn't mapping the nucleotide sequences allow for the detection of chimeric reads? multi-mappers or reads with multiple primary alignments with high mapping score would indicate this. This ideally would be done first, to filter the read IDs which give full length sequences, of which to then remove headers, and potentially malformed tail sequences (not currently assessed in this analysis). Then the resultant squiggles used for DTWA, as they have the highest chance of matching the gold standard.

Doing everything in squiggle space only seems to miss the advantage of having sequence information available, especially if the goal is to do modification detection in the future. The only reason to need to do everything in signal space would be if it was being used for readUntil/adaptive sequencing, which based on the methods, was not considered (using the tail end for setting mean for leader trimming, using matlab)

2.2 DTWA Algorithms and proposed success measures

195: The OEM provided nucleotide-to-picoamperage mapping should be named, and provided, and referred to here.

249-254: if there is a limitation of "inserting events", why not go back to the raw signal where the events were generated from? If a particular event was generated from a rather large section of raw signal, the number of inserted events could be estimated, or an orthogonal segmentation method from any number of open source tools could be used to attempt to insert the correct events.

Results:

Figure 2. The blue box showing similarity of signals, is not entirely clear given the scale of the figure. A better representation fo this would be to zoom that to a full supplementary figure, colour each squiggle differently, and overlap the plots. This can get messy, so perhaps only half of the selected squiggles need to be used. As it stands, I can't tell any similarity.

3.2.1 Enolase squiggle investigation

331-332: Is this code provided? if so, which files. If not, please include, along with how the timing was measured for all methods. There is no way to reproduce the timings without this.

3.2.3 Qualitative analysis of the DTWA consensus characteristics.

378:380. The x1.7 longer squiggles, as stated by the authors previously, has not been checked against other datasets. It is also contingent on the older RNA001 dRNA kit (RNA002 is the current kit), and the various cleaning metrics rely on the segmentation method used, which is currently unknown due to the lack of information or raw data from the authors.

The gold standard is also unknown, as the model used to produce it is not mentioned or provided. How this model is then converted into the gold standard text files with picoamp event data points, is not provided, and thus unclear.

Figure 4 (and most of the comparisons between time warped squiggles): When comparing time warped squiggles. Stacking squiggles is a rather clunky way of demonstrating the similarities/differences between them. Overlapping signals is one way (though still quite messy). Plotting the dtw paths, especially plotting each method's consensus vs gold paths, would not only give a measure of distance, but also if the methods are consistent across the various warped regions of the reads. It should be much clearer to assess the various approaches and demonstrate the various features of each method. (are certain regions handled better/worse for each method?/Is there limitations to how much warping can be corrected?/is there a better method other than global scoring?/etc)

I have added an image as an example of plotting the warping path on top of a dtw distance score matrix (as a heatmap) to demonstrate warping features between 2 squiggles of the same sequence to give an idea of what I mean.

You may also consider using something other than matlab to plot this data, as the quality of matlab plots is quite poor. Python or R have libraries with a reputation for producing clearer plots for publication, and are also free.

5. Conclusion

562: Identification of a potential chemical nucleotide modification would be relatively straight forward to demonstrate using the current method, by analysing any of the number of dRNA modification datasets now published, with modified and unmodified controls. producing the consensus against the gold standard for untreated, then compare the treated reads against it. By comparing the warp paths and mapping that back to the associated base, a measure of modification can be deduced. This is indeed an exciting path forward.

Spelling/grammar

100: identity -> identify

131: enblign -> enabling (i think?)

182: averaged -> average

538: nanosequencer -> nanopore sequencer

538: nanostream - > datastream/signal

Reviewer #2: Overall this is a well-written paper that I think will benefit from making the technical methods clearer. I think other reviewers may be more qualified to comment on the methodology of the data generation and new evaluation metrics. But I think this paper reads well and, to my knowledge, represents a novel contribution to the field.

- If I understand the paper correctly, voting is used to improve a single DTWA algorithm; in particular, it is NOT used to combine the results of several DTWA algorithms. Is this correct? If so, I think this should be made crystal clear; it took me a couple of readings to understand this.

- Line 187 - line number is in the middle of the equation This is a problem for all of the equations

- Section 2.2: I think this paper would benefit greatly from giving more background on how a consensus is generated. I assume these three algorithms take the noisy squiggles as input, compute a consensus squiggle that optimizes a particular objective function. I think this should be explained; and in doing so, foreshadow what is missing from these

algorithms that voting can fix.

- Line 186: it says that the goal is to minimize the mean and standard deviation of the DTW distance. Immediately following is equation (1). This wording is somewhat confusiong; equation (1) shows how to compute a single number, so what would be the mean and standard deviation of this sone number? My guess is that equation (1) is the formula for the mean that is being minimized; if this is correct, consider adding another equation with the formula of the standard deviation being minimized. If my interpretation is incorrect, then this paragraph needs more explanation of what you are taking the mean/standard deviation of when measuring the success of an algorithm.

- 198: "3 x 3 = 9 datasets" To be clear: this means that for each of the 3 algorithms, and each of the 3 datasets, you applied the algorithm to that dataset, to generate 9 different consensus sequences, correct?

- Line 201: It is unclear to me what you mean when you say "comparing DTWdist_{gold -> ensemble} against DTWdist_{consensus->ensemble}". In a "good" consensus, should the two metrics be similar? Should the consensus be much closer to gold than to the ensemble?

- Line 206: What is DTWdist with no subscript? I was onder the impression that the DTWdist metric should have a subscript delineating what two sequences (and/or sequence ensembles) are being compared. Perhaps DTWdist = DTWdist_{gold -> ensemble} - DTWdist_{consensus->ensemble}? If so make this explicit.

- Lines 238-245: Please explicitly define the measures you used to evaluate the results of voting

- Line 339: font size for "nDTWdist_{gold -> ensemble}" is off

- Line 349: font size for "nDTWdist_{consensus -> ensemble}" is off

- Line 461: This is the standard deviation across 7 different consensus signals (before after voting) - correct? If so please make this explicit.

- Results and (especially) conclusion are well-written

Reviewer #3: The authors propose an optimization application supported by theory, with illustrative validations. Comments:

1. Please edit the paper carefully such that to respect the instructions for authors. A homogeneous style is desired.

2. You should present the contributions with respect to your past papers that should be cited. Your past algorithms are very well appreciated.

3. The optimization problem is not defined. You are speaking several times about optimization and also including optimization algorithms but an overall optimization problem is not given.

4. As mentioned in the comment 3, the definitions of the optimization problems must be treated with attention. The authors are advised to include the following representative applications of optimization problems and algorithms as they are successful in various fields: Gene finding in the chicken genome (BMC Bioinform 2005), Second order intelligent proportional-integral fuzzy control of twin rotor aerodynamic systems (PCS 2018), Whale optimization algorithm for performance improvement of silicon-on-insulator FinFETs (IJAI 2020).

5. The connection between the optimization algorithms and the optimization problem is also not pointed out.

6. The connection between the application section and the previous theory is not clear enough. More details are necessary for improved transparency.

7. You should save the code of programs and examples, and cite the link to them in the paper. This is useful for validation, and helps the above comment. The importance of this comment is related to the fact that similar optimization algorithms are reported in the literature, they report excellent results but cannot be tested.

8. You should specify which are the parameters of the optimization algorithms, which of them should be selected by the user and which of them are random.

9. I am not sure if the comparison is correct because all algorithms used in the comparison including yours depend on parameters. Other parameters lead to other results.

10. The stochastic effects are not reflected in the results.

Concluding, the paper has a strong potential for being appreciated and cited, but it requires improvements and also extension.

**Have the authors made all data and (if applicable) computational code underlying the findings in their manuscript fully available?**

Reviewer #1: **No: **The fast5 files, containing the raw signal data which the event values (mean value from signal segmentation) are not available. The code provided is from a different publication (Evaluation of Simulation Models to Mimic the Distortions introduced into Squiggles by Nanosequencers and Segmentation Algorithms) and is written in matlab (software requiring a rather expensive paid license). The code for this paper is not provided, and not maintained in an open repository, such as github/gitlab/etc.

Reviewer #2: Yes

Reviewer #3: **No: **

PLOS authors have the option to publish the peer review history of their article (what does this mean?). If published, this will include your full peer review and any attached files.

Reviewer #1: **Yes: **James M. Ferguson

Reviewer #2: **Yes: **Arjun Chandrasekhar

Reviewer #3: No
---

## [Decision Letter · Decision Letter 1]

15 Aug 2021

Dear Dr. Smith,

We are pleased to inform you that your manuscript 'Evaluating the effectiveness of ensemble voting in improving the accuracy of consensus signals produced by various DTWA algorithms from step-current signals generated during nanopore sequencing' has been provisionally accepted for publication in PLOS Computational Biology.

Before your manuscript can be formally accepted you will need to complete some formatting changes, which you will receive in a follow up email. A member of our team will be in touch with a set of requests. Please accommodate the last corrections of the Reviewer 1 on the same occasion.

Best regards,

Eduardo Eyras, PhD

Guest Editor

PLOS Computational Biology

Ilya Ioshikhes

Deputy Editor

PLOS Computational Biology

Reviewer's Responses to Questions

**Comments to the Authors:**

Reviewer #1: I would like the thank the authors for their detailed and thorough response to my initial review of the work. The work is now much clearer, and with the additional method descriptions, version information, and scripts provided in the github repo, the work is also now reproducible.

The authors have answered my questions to a satisfactory level.

As a comment to the authors, in relation to the last paragraph before the conclusion, hypothesising about z-normalizing each signal. In my own research, I found median median-absolute-distance (med-MAD) to be a far superior method of normalisation than z-scaling, at least for RAW signals, even after pA conversion. I would imagine this would transfer well to event segmented data too. An example python code snippet is provided below

# sig is a an int numpy array, with any excessivley large spike datapoints removed

arr = np.ma.array(sig).compressed()

med = np.median(arr)

mad = np.median(np.abs(arr - med))

scaled_mad = mad * 1.4826

mad_sig = []

for i in sig:

mad_sig.append((i - med) / scaled_mad)

sig = np.array(mad_sig)

All the best with your future work.

James Ferguson

Reviewer #2: All my comments have been dealt with in a satisfactory manner.

Reviewer #3: The paper is improved and can be published.

**Have the authors made all data and (if applicable) computational code underlying the findings in their manuscript fully available?**

Reviewer #1: Yes

Reviewer #2: Yes

Reviewer #3: Yes

PLOS authors have the option to publish the peer review history of their article (what does this mean?). If published, this will include your full peer review and any attached files.

Reviewer #1: **Yes: **James M. Ferguson

Reviewer #2: **Yes: **Arjun Chandrasekhar

Reviewer #3: No

---

## [Editor Report · Acceptance letter]

23 Aug 2021

PCOMPBIOL-D-21-00540R1 

Evaluating the effectiveness of ensemble voting in improving the accuracy of consensus signals produced by various DTWA algorithms from step-current signals generated during nanopore sequencing

Dear Dr Smith,

I am pleased to inform you that your manuscript has been formally accepted for publication in PLOS Computational Biology. Your manuscript is now with our production department and you will be notified of the publication date in due course.

With kind regards,

Zsofi Zombor
